# Risk Factors Associated with the Alpine Multispecies Farming System in the Eradication of CAEV in South Tyrol, Italy

**DOI:** 10.3390/v13101959

**Published:** 2021-09-29

**Authors:** Alexander Tavella, Katia Capello, Giuseppe Bertoni, Astrid Bettini

**Affiliations:** 1Laboratory for Serology and Technical Assistance, Istituto Zooprofilattico Sperimentale delle Venezie, 39100 Bolzano, Italy; atavella@izsvenezie.it; 2Office for Analytic Epidemiology and Biostatistics, Istituto Zooprofilattico Sperimentale delle Venezie, 35020 Legnaro (PD), Italy; kcapello@izsvenezie.it; 3Institute for Virology and Immunology, Vetsuisse Faculty, University of Bern, 3012 Bern, Switzerland; bertoni@vetsuisse.unibe.ch

**Keywords:** CAEV, SRLV, multispecies farming system, cross infections

## Abstract

South Tyrol has implemented, in 2007, a mandatory eradication program against Caprine Arthritis Encephalitis Virus (CAEV), a virus known to cause economic losses related to decreases in milk production and milk quality in goats, along with poor animal welfare and premature death. After a great initial decrease in the seroprevalence, the program has reached a tailing phase with scattered positivities. Potential risk factors associated with the multispecies farming system, a traditional approach in South Tyrol, are evaluated in this study, in order to better understand some of the potential causes leading to the tailing phenomenon. A statistically significant number of farms was selected for the present study, based on the risk factors evaluated. Even though there is no statistically significant association between the practices evaluated and the incidence of infection, the authors believe that it is important to highlight potential risks that may threaten the outcome of this eradication program.

## 1. Introduction

Small Ruminant Lentiviruses (SRLVs) are viruses of the *Retroviridae* family that include Caprine Arthritis Encephalitis Virus (CAEV) and Maedi-Visna Virus (MVV) [1]. SRLVs can cause chronic and progressive inflammatory and degenerative lesions to the joints, mammary glands, lungs and central nervous system in both goats and sheep [2,3,4]. Many economic losses have been attributed to SRLV infections, such as a decrease in milk production and milk quality, along with poor animal welfare and premature death [2,4,5,6,7]. SRLVs have been characterized into five genotypes (A–E), three of which (A, B and E) have been further divided into viral subtypes (A1-22, B1-5, E1-2) [8,9,10]. SRLV B, in particular subtype B1, is considered the prototype for CAEV, while SRLV A is the prototype for MVV [2,11]. SRLVs are promiscuous and readily cross the species barrier between goats and sheep [2,8]. There is no evidence of a strict specialization of particular SRLV genotypes to goats or sheep, with the exception of genotype E, which appears to infect only goats [12]. In contrast, the factors regulating viral transmission efficiency appear to be slightly different in the two species. Vertical transmission through ingestion of infected colostrum is pivotal to sustain a chain of infection in goats, while horizontal transmission plays a prominent role in MVV infections in sheep [11,13].

The Autonomous Province of Bolzano, South Tyrol (Italy), has implemented, in 2007, a compulsory eradication program against CAEV that foresees the serological investigation of anti-SRLV antibodies and the detection of the infecting genotype/subtype in all goats older than 6 months of age, with the culling of all SRLV B-infected goats [14,15]. No goats infected with other viral genotypes are subjected to culling. Sheep are, by decree, not sampled during the eradication program. The exception to this rule resides in multispecies farms, which are farms that simultaneously breed both goats and sheep along with other species: in the presence of seropositive goats, sheep are mandatorily subjected to SRLV screening as well [14]. In these farms, all sheep older than 6 months are tested for anti-SRLV antibodies and subjected to indirect genotyping, but the culling of sheep is not mandatory [16]. Even though sheep are not considered part of the local eradication program, several studies have been conducted since the beginning of the plan, in order to evaluate the SRLV seroprevalence in sheep and to investigate possible risk factors associated with the presence of sheep in the local multispecies farms [17,18].

In this work, the authors investigated the risk factors potentially associated with the alpine multispecies farming system. The investigated risk factors were: the traditional practice of summer alpine pasture grazing, the type of facility in which sheep and goats are kept within multispecies farms and international trade of live animals. The aim of this study was to evaluate the epidemiological role of sheep as a reservoir for SRLVs, to optimize the current eradication program based on the newly acquired scientific information and to develop general guidelines applicable in other regions with similar geographical characteristics and animal husbandry (traditional multispecies farming systems in the Alps and Appenine Mountains).

## 2. Materials and Methods

### 2.1. Study Design

All goats older than 6 months are subjected to the yearly CAEV prevention campaign [16]. To obtain information on the SRLV situation in sheep, we relied on sheep sera collected in the frame of the mandatory control programs for *Brucella melitensis* and *Brucella abortus,* and for rams, for *Brucella ovis* [19]. The current small ruminant population counts 26,806 goats and 42,976 sheep [20]. These data, along with the data from the 2016–2017 CAEV prevention campaign, were evaluated in order to achieve a statistically significant number of farms and animals participating in the present study.

During the prevention campaign under study, blood samples of all animals belonging to 51 multispecies farms with previous SRLV-positive or dubious serological results (from now on referred to as “non-negative”), 57 negative multispecies farms and 93 sheep-monospecies farms were analyzed, and the resulting data were recorded. Blood serum samples of all sheep belonging to these farms were collected and subjected to serological investigations. A more detailed overview of the sampled farms is shown in Table 1.

### 2.2. Serological Analyses

Blood samples were drawn from the jugular vein by means of vacuum blood collection tubes with a clotting activator (Vacutest Kima, Arzergrande, Italy), and centrifuged at 1646 *g* for 3 min to obtain the serum. As a first screening, the sera of all goats and sheep of the selected farms were tested for anti-SRLV antibodies with a commercial kit “Id.Vet” (ID Screen^®^ MVV/CAEV Indirect Screening Test, ID.Vet Innovative Diagnostics, Grables, France). All sera belonging to farms with at least one non-negative animal were tested with a second commercial screening kit “Eradikit Screening” for the detection of anti-SRLV antibodies (Eradikit^TM^ SRLV Screening Kit, IN3 Diagnostics, Torino, Italy). All procedures were performed according to manufacturers’ instructions.

### 2.3. Statistical Analysis

Several risk factors were evaluated based on the serological data: presence/absence of sheep within a non-negative farm, alpine pasture during the summer season, type of farming system and international trade of animals to other European countries. Regarding multispecies farms, the chi-square test was adopted to evaluate the association between risk factors and the presence of infected individuals. The same statistical test was used for the association between the status of farms (negative vs. non-negative) and the presence or not of sheep within the farm. The non-parametric Wilcoxon rank-sum test was applied to test possible differences in the number of sheep between non-negative and negative multispecies farms. The SRLV antibodies results for multispecies farms were summarized calculating the proportion of non-negative farms over tested farms (i.e., seroprevalence with 95% confidence interval); specifically, for non-negative farms, the mean and median of the proportions of non-negative animals were provided (i.e., intra-farm prevalence). All statistical analyses were performed using Stata v. 12.1.

## 3. Results

A total of 201 farms were sampled for this study: 108 were multispecies and 93 were sheep monospecies farms. Three aspects were evaluated as possible risk factors: (i) the traditional practice of seasonal pasture grazing, (ii) the presence of a separate facility for goats and sheep or a single facility in which both species are held and (iii) the trade of live animals with foreign Countries (mainly Austria, Germany, Switzerland and the Netherlands).

As a preliminary step, we investigated the distribution of non-negative farms by evaluating the presence or absence of sheep within the farm (Table 2). A significantly higher percentage of positive farms was highlighted in the case of presence of sheep (*p*-value 0.0008). On the other hand, approximately equal distribution between presence and absence of sheep was found in the negative farms.

Furthermore, the distribution of the number of sheep present in non-negative multispecies farms and negative multispecies farms was evaluated (Figure 1). The consistency of the multispecies farms in terms of number of sheep present within the farm itself resulted not statistically different between negative and non-negative multispecies farms.

For the purposes of this study, all sheep belonging to the selected multispecies farms were tested for anti-SRLV antibodies, using the same commercial kit used for the first screening analysis (Id.Vet). Overall, 41 of the 51 non-negative multispecies farms and 52 of the 57 negative multispecies farms were tested. Seven (17%) SRLV-non-negative farms and fifteen (29%) CAEV-negative farms had at least one positive sheep. All data are reported in Table 3.

Of the 93 sheep monospecies farms, 27 presented at least 1 anti-SRLV non-negative animal. Furthermore, the distribution of the number of sheep present within each farm was analyzed and the results are presented in Table 4.

The first potential risk factor evaluated within this study was the habit of seasonal alpine pasture grazing practiced by many breeders in South Tyrol. All data are presented in Table 5. Out of the 108 multispecies farms tested, 49 farms remain in the same territory, while 59 farmers take their animals to alpine pastures. In more detail, 27 non-negative multispecies farms go to alpine pastures, while 24 non-negative multispecies farms do not practice this habit. On the other hand, of the 93 sheep monospecies farms, 41 practice alpine pasture grazing, while 52 stay in their original territory throughout the year. There is no statistically significant association between this practice and the incidence of infection (*p*-value = 0.739).

The second potential risk factor evaluated was the presence of different facilities to separate goats from sheep within the same farm or the presence of a single facility in which both species are kept together. Out of the non-negative multispecies farms, 20 have separate facilities and 31 keep their animals together, while out of the negative farms, 25 have separate facilities and 32 have one single facility (Table 6). There is no statistically significant association between the type of facility in which the two species are held within a multispecies farm and the presence of infected individuals (*p*-value = 0.625).

The third and last potential risk factor evaluated in this study was the import history of the farms with foreign countries. Of the 51 multispecies non-negative farms, 5 trade their animals with other countries, while out of the 57 multispecies negative farms, only 1 farm imports animals from foreign countries. Finally, out of the 93 sheep monospecies farms, only 5 have imported animals from other European countries (Table 7). In this particular case, no statistical test was run because of the low number of samples that practice international trade.

## 4. Discussion

An initial significant decrease in the seroprevalence of SRLV in goats has characterized the mandatory eradication program of the Autonomous Province of Bolzano since its beginning; however, in the past few years, a tailing phenomenon has been observed, which is particularly bothersome for the plan’s management [14,15]. The same tailing phenomenon has been observed in several other eradication and control programs worldwide, such as the Swiss program, and appears to be correlated not only with the diagnostic differences among the different plans, but mainly with the complex biology of the virus itself [11,21]. The genetic variability, along with the absence of universal diagnostic protocols able to identify all possible infecting genotypes and viral subtypes, represents an important limitation to the diagnostic measures of SRLV infections [12,13,22]. Furthermore, interspecies transmission between sheep and goats seems to be extremely important in farms where both species are bred together [2,23,24,25] and in close contact. This specific aspect is particularly evident in South Tyrol, where 30–33% of the total amount of farms are characterized by a multispecies farming system [20]. The current SRLV eradication program does not foresee the serological investigation of sheep, and substantial data on the seroprevalence of the ovine population in South Tyrol are therefore not available.

In 2007 and again in 2012, two pilot studies were conducted in order to achieve information on the SRLV seroprevalence in sheep. Briefly, in 2007, 1117 goat monospecies farms, 1423 sheep monospecies farms and 741 multispecies farms were sampled. Among the goat monospecies farms, 29.7% presented seropositive goats, while among the sheep monospecies farms, 4.4% had seropositive sheep. Furthermore, among the multispecies farms, 2.7% presented both seropositive goats and sheep, 34.1% had only seropositive goats and 2.4% presented only seropositive sheep. In the 2007–2008 prevention campaign, 1.9% of the 7627 tested sheep resulted non-negative. Similarly, in the 2012–2013 prevention campaign, 1.67% of the 7513 tested sheep resulted non-negative. When comparing these data with the results of the data collected in the present study, a seroprevalence of 4.25% was calculated, which compared to the approximately 1% seroprevalence in the goat population measured during the previous studies is a considerably high value [17,18].

The data presented in this study report that the risk factors evaluated must be further analyzed and taken into consideration, because they may produce a significant interference in the serology of SRLVs when associated with multispecies farming systems. The identification of such risk factors represents an important turning point in understanding the virus dynamics occurring in the last phase of an eradication program. The presence within the same farm of both goats and sheep, between which viral transmission has been widely demonstrated, may give rise to well-founded suspicion on the role of this species as a viral reservoir. In the present study, the presence of sheep in multispecies farms does not represent a significant risk factor, however, it serves as a boosting factor in the reaction intensity of seropositive samples [15].

The first evaluated risk factor was the common practice of summer alpine pastures typically adopted by many of the farms under study. This practice could potentially lead to viral transmission if seropositive animals were to be present in these locations. Alpine pastures are a very extensive farming practice, with a relatively low risk of viral transmission between animals. Sheep and goats do not normally mix paths while on alpine pastures. The main risk of contact is therefore when drinking water out of common drinking areas or licking salt off of common rocks. Thanks to the control measures adopted by decree in the local eradication program [14], only seronegative goats are allowed on alpine pastures, and the risk of transmission is therefore greatly decreased. It is noteworthy that, in the presence of false-negative serological results or late seroconversion, a particular animal could be allowed on the alpine pasture and unintentionally spread the virus through horizontal transmission. Furthermore, sheep are not tested for anti-SRLV antibodies, as provided for by decree. This means that all sheep belonging to negative multispecies farms are sent to alpine pastures with an unknown SRLV sanitary status and, if non-negative, can potentially infect individuals belonging to other farms. This aspect must be therefore taken into consideration and continuously monitored. Fortunately, this event does not seem to be happening often, and the evaluation of this risk factor is not statistically significant.

The type of farming system in which goats and sheep are bred was evaluated as well. Multispecies farms can present either completely separate buildings in which the different species are kept, or a single building where sheep and goats live together. This aspect is greatly associated with the morphological conformation of the South Tyrolean landscape, which is characterized by an alpine to sub-alpine territory, where large separate buildings are frequently impossible to build. Keeping sheep and goats in the same facility may lead to easier viral transmission through nasal discharge and aerosol. However, this does not seem to be statistically affecting the prevalence of the virus in South Tyrol.

Last but not least, the import of animals from foreign countries was evaluated as a risk factor potentially affecting viral transmission. Many farms in South Tyrol practice internal breeding, without ever using other farms’ bucks or buying kids from other farms. This practice is theoretically the best method for avoiding the introduction of positive animals in a naïve farm. Other farmers acquire animals from farms with the same sanitary status as them, in this way maintaining the same sanitary status as before. Finally, as South Tyrol is the northern-most Province of Italy and borders foreign countries, several farms practice international trade, mainly buying animals from Austria, Germany, Switzerland and the Netherlands. For foreign import, a sanitary certificate indicating the health status of the animals regarding Brucella is acquired. Unfortunately, the use of different diagnostic tools used by the different laboratories for the detection of SRLV antibodies may lead to the acquirement of false-negative animals. As very few farms import animals from other European countries, no statistical analysis was conducted. However, it seems fair to state that foreign import does not represent a statistically significant risk factor.

The results of this study highlight the good performances achieved in the CAEV eradication program of the Autonomous Province of Bolzano. In more detail, the initial phase was characterized by a drastic decrease of the seroprevalence with a complete elimination of clinical signs in goats, followed by a second phase characterized by a tailing phenomenon presenting scattered positivities among the Province [14]. This tailing phenomenon can be related to different causes, such as (i) false-positive reactions, (ii) false-negative reactions in the previous prevention campaigns and (iii) the close contact between sheep and goats and the potential transmission in multispecies farming systems [15]. The potential risk factors evaluated in this work do not present a statistically significant association with the multispecies farming system. Even though these results seem promising, a continuous monitoring of these and other potential risk factors remains of utmost importance.

## Figures and Tables

**Figure 1 viruses-13-01959-f001:**
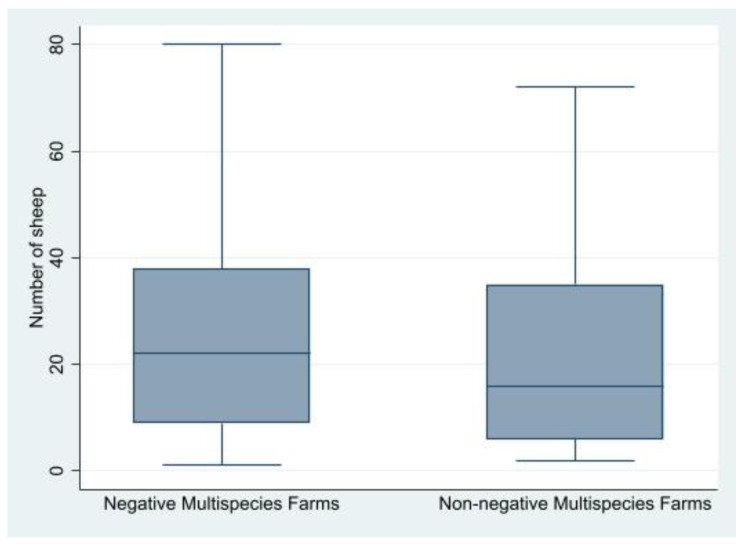
Distribution of the number of sheep within non-negative and negative multispecies farms.

**Table 1 viruses-13-01959-t001:** Descriptive statistics of the studied populations.

Type of Farm	Number of Farms	Total Number of Goats	Mean Number of Goats within Farm	Total Number of Sheep	Mean Number of Sheep within Farm
Multispecies Negative Farm	57	1280	22.46	1651	28.96
Multispecies Positive Farm	51	1796	35.22	1614	31.65
Sheep Monospecies Farm	93	n.a.	n.a.	972	10.45

**Table 2 viruses-13-01959-t002:** Distribution of negative and non-negative farms by presence of sheep within the farm.

	Only Sheep	Presence of Sheep	Total
Non-negative Farms	27 (34.6%)	51 (65.4%)	78 (100%)
Negative Farms	66 (53.7%)	57 (46.3%)	123 (100%)
Total	93 (46.3%)	108 (53.7%)	201 (100%)

**Table 3 viruses-13-01959-t003:** Descriptive statistics of SRLV results for sheep belonging to multispecies farms.

	Tested Farms	SRLV Non-Negative Farms	Seroprevalence (with 95% CI)	Intra-Farm Mean Prevalence
Non-Negative Multispecies Farms (*n* = 51)	41	7	17% (95% CI: 7%–32%)	12% (Median = 7%)
Negative Multispecies Farms (*n* = 57)	52	15	29% (95% CI: 17%–43%)	12% (Median = 5%)

**Table 4 viruses-13-01959-t004:** Descriptive statistics of SRLV results for the 93 sheep monospecies farms.

	SRLV Non-Negative Farms	Intra-Farm Mean Prevalence	Mean Number of Sheep/Farm	Standard Deviation	Median Number of Sheep	1st Quartile	3rd Quartile
Sheep Monospecies Farms (*n* = 93)	27	25% (Median = 12.5%)	10	14	8	4	11

**Table 5 viruses-13-01959-t005:** Risk factor 1: Results of the association between the traditional practice of seasonal pasture grazing and type of farm.

Pasture Grazing	Multispecies Non-Negative Farms	Multispecies Negative Farms	Total Multispecies Farms	Sheep Monospecies Farms
Yes	27 (52.94%)	32 (56.14%)	59	41 (44.1%)
No	24 (47.06%)	25 (43.86%)	49	52 (55.9%)
Total	51	57	108	93

**Table 6 viruses-13-01959-t006:** Risk factor 2: Results of the association between type of facility adopted and type of farm.

Type of Facility	Multispecies Non-Negative Farms	Multispecies Negative Farms	Total
Separate	20 (39.22%)	25 (43.86%)	45
Single	31 (60.78%)	32 (56.14%)	63
Total	51	57	108

**Table 7 viruses-13-01959-t007:** Risk factor 3: Results of the association between import history and type of farm.

Trade	Multispecies Non-Negative Farms	Multispecies Negative Farms	Total Multispecies Farms	Sheep Monospecies Farms
No	46 (45.10%)	56 (54.90%)	102	88 (94.6%)
Yes	5 (83.33%)	1 (16.67%)	6	5 (5.4%)
Total	51	57	108	93

## Data Availability

All data is provided in the manuscript.

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
