# Peer review of "Risk Factors Associated with the Alpine Multispecies Farming System in the Eradication of CAEV in South Tyrol, Italy"

_viruses, 2021, doi:10.3390/v13101959_

Round 1

Reviewer 1 Report

In this work the authors investigated the potential risk factors potentially associated with the alpine multispecies farming system. Only three risk factors were evaluated which appeared to be not statistically significant.  The manuscript does not bring any original/significant  findings to the SRLVs field whats why is not interesting enough for "Viruses".

Introduction

  • L30- 22 subtypes were detected in group A. Please update data and references.
  • L34- remove “anti-CAEV”. ELISA tests detect antibodies against CAEV and MVV. Due to the occurrence of interspecies infections, it is not known with what genotype the goats and sheep are infected. This can only be confirmed by sequencing.
  • L35-“ with the culling of all SRLV B infected goats” What in the case when goats are infected with a genotype other than B?
  • L37- what mean multispecies farms? Sheep+ goats?
  • L38- SRLV or MVV/CAEV instead of anti-MVV
  • L39-“SRLV B-positive sheep” Do you mean sheep infected by CAEV-like viruses?

Materials and methods

  • The entire section “study design” should be corrected because its incomprehensible. It should be clearly described from which source the samples come and what how many samples were tested (animals and flocks). It is not known whether samples are newly collected or described in the cited publications. How many animals is in the flocks? It should be explained what means non-negative farms? I suggest change this term
  • L74 serologically positive animals instead non-negative animals
  • for serological analysis, genotyping tests (Eradikit genotyping and “SU5” ELISAs) were used but in the results there is no information about genotypes

Results

  • L95-98 this is the purpose of the study, which is repeated again
  • Description of the table 1 is not appropriate because the table includes data for negative and "positive" flocks not only for non-negative.
  • Also descriptions of table 2, 4 and 5 are inadequate and must be corrected
  • Why not all 108 multispecies farms were analyzed? How the farms were selected? Why the Samples from sheep were tested again? Did the samples get resampled for testing?
  • Using the terms CAEV-farms and MVV-farms is a bit confusing. It would be good not to use these names.
  • How the prevalence was calculated?
  • L125 These results have already been presented in Table 1 as preliminary step
  • Why the analysis of distribution of the number of sheep within each farm was done only for monospecies farms?
  • L134 108 or 93 farms were analyzed?

Dicussion

  • L183-198 the information presents data from the prevention campaign without relating to the results obtained in the work
  • L201 in the introduction and in results L95-L98 only three risk factors are mentioned
  • There is a lot of literature on SRLV risk factors but there is no comparison to any literature in the discussion.

Author Response

Introduction

  • L30- 22 subtypes were detected in group A. Please update data and references. Revision accepted.
  • L34- remove “anti-CAEV”. ELISA tests detect antibodies against CAEV and MVV. Due to the occurrence of interspecies infections, it is not known with what genotype the goats and sheep are infected. This can only be confirmed by sequencing. Revision accepted.
  • L35-“ with the culling of all SRLV B infected goats” What in the case when goats are infected with a genotype other than B? A sentence about goats infected with viral genotypes other than SRLV B not being subjected to mandatory culling was added. These animals, even though infected with SRLV, stay in their flock.
  • L37- what mean multispecies farms? Sheep+ goats? The term "multispecies farms" was explained within the manuscript (yes multispecies = sheep + goats).
  • L38- SRLV or MVV/CAEV instead of anti-MVV Revision accepted.
  • L39-“SRLV B-positive sheep” Do you mean sheep infected by CAEV-like viruses? Yes, the sentence was revised in order to avoid misunderstantings.

Materials and methods

  • The entire section “study design” should be corrected because its incomprehensible. It should be clearly described from which source the samples come. The source of the samples was better described within the text – briefly: goat samples were collected within the CAEV eradication program, while the information on sheep SRLV status was obtained by performing SRLV analyses on samples collected within the Brucella monitoring programs. and what how many samples were tested (animals and flocks) The requested data was added. It is not known whether samples are newly collected or described in the cited publications. The samples are newly collected and not described in any other publication (citations are referred to the decree, not to publications). How many animals is in the flocks? This data was added to the manuscript. It should be explained what means non-negative farms? Revision accepted. I suggest change this term The term was explained. The authors believe it is a better term to describe both positive and dubious results (as the ELISA kits foresee both positive and dubious results), rather than just stating that they are all positive.
  • L74 serologically positive animals instead non-negative animals The term “non-negative” was explained within the text. The authors believe this term is a better fit for the concept.
  • for serological analysis, genotyping tests (Eradikit genotyping and “SU5” ELISAs) were used but in the results there is no information about genotypes This information was added to describe the process to which all samples are subjected. The data was not relevant for the purposes of this study and was therefore not included in the result section. A sentence explaining this was inserted in the manuscript as well.

Results

  • L95-98 this is the purpose of the study, which is repeated again The aim of the study was stated again to introduce the results’ section.
  • Description of the table 1 is not appropriate because the table includes data for negative and "positive" flocks not only for non-negative. Non-negative means positive and dubious – this definition was included in the manuscript to clarify the concept. The table description was modified.
  • Also descriptions of table 2, 4 and 5 are inadequate and must be corrected Revision accepted - The descriptions of tables 2,4 and 5 were modified.
  • Why not all 108 multispecies farms were analyzed? 108 multispecies farms were analyzed: this data consists of 51 SRLV positive farms and 57 SRLV negative farms. A table was added in the materials and methods section to have a better overview of the tested farms. How the farms were selected? 51 multispecies farms with a history of SRLV positivity + 57 multispecies farms with a history of SRLV negative results + 93 farms that hold only sheep were selected in the present study (data described in the Material and Methods section). Why the Samples from sheep were tested again? Sheep samples were not tested again. Did the samples get resampled for testing? The samples were collected once within the SRLV eradication program (goats) and the Brucella monitoring program (sheep) – samples were collected only once and multiple serological analyses were performed.
  • Using the terms CAEV-farms and MVV-farms is a bit confusing. It would be good not to use these names. Revision accepted
  • How the prevalence was calculated? At farm level, the prevalence was calculated as number of positive farms for SRLV over the total tested farms for SRLV; the intra-farm prevalence was the number of positive animals over the total number of animals tested in the farm, using only farms tested positives to SRLV. These information were added in the Materials and Methods' section.
  • L125 These results have already been presented in Table 1 as preliminary step Table 1 only reports the number of non-negative farms, while table 3 shows a richer statistical analysis of the 93 sheep farms.
  • Why the analysis of distribution of the number of sheep within each farm was done only for monospecies farms? The present study was initially set on the evaluation of multispecies farms only, the 93 sheep-monospecies farms were included in the study as a “control group”.
  • L134 108 or 93 farms were analyzed? The evaluation of risk factors was performed on the 108 (51+57) multispecies farms.

Dicussion

  • L183-198 the information presents data from the prevention campaign without relating to the results obtained in the work. This data was included in the discussion section, as it describes data previously collected (2007-2008 prevention campaign) when a study on SRLV prevalence in sheep was conducted. This data was added to provide a little more background on the situation in the sheep population in South Tyrol.
  • L201 in the introduction and in results L95-L98 only three risk factors are mentioned. Thank you for the observation, this is a mistake from a previous draft. The manuscript was modified accordingly.
  • There is a lot of literature on SRLV risk factors but there is no comparison to any literature in the discussion. The aim of this study was to evaluate risk factors related to the South Tyrolean reality; literature on studies evaluating risk factors (some of which not applicable to the South Tyrolean situation) was provided to support our work, but no comparison was made because it did not fit with the aim of the study.

Reviewer 2 Report

No particular advice is given to authors except to review minimal spelling errors

Author Response

The authors kindly thank the review for the time and availability. The manuscript was revised in order to correct all spelling errors.

Reviewer 3 Report

An interesting, well-written manuscript describing possible risk factors associated with CAEV in South Tyrol - Italy.

It would be useful to include some information in the introduction describing the difference between sheep and goats in risk factors associated with SRLV infection. Some description of the virus genotypes would also be useful.

In the Materials and Methods, it states that the SRLV infecting genotypes are tested as well as the viral sub-type, yet the information is not included in the results. Is there a primary genotype or subtype being identified?

Since it is stated that SRLV can be transmitted between sheep and goat, can you include some information why the eradication plan is for goats only and sheep are not being tested for SRLV?

P5, L156: suggest changing "trade" to "import since for the subject of this manuscript, animals leaving the country are not important.

Regarding transmission between sheep and goats - on the multi-species farms, do the animals lamb and kid together? This may be where transmission occurs. 

Author Response

It would be useful to include some information in the introduction describing the difference between sheep and goats in risk factors associated with SRLV infection. Some description of the virus genotypes would also be useful. A brief description was added win the introduction section.

In the Materials and Methods, it states that the SRLV infecting genotypes are tested as well as the viral sub-type, yet the information is not included in the results. Is there a primary genotype or subtype being identified? This information was added to describe the process to which all samples are subjected. The data was not relevant for the purposes of this study and was therefore not included in the result section. A sentence explaining this was inserted in the manuscript as well.

Since it is stated that SRLV can be transmitted between sheep and goat, can you include some information why the eradication plan is for goats only and sheep are not being tested for SRLV? Initially, the CAEV eradication was requested by the local Breeding Association. The Veterinary Service implemented the program on goats only because, based on previous studies, the prevalence in goats was higher in goats than in sheep. Furthermore, goats are economically more relevant, as they are used for the production of milk for caprine dairy products, while sheep are bred mainly for hobby reasons. Since part of the request for an eradication program was based on achieving an SRLV-free sanitary status in order to be able to improve the caprine dairy products’ market, the local authorities, in accordance with the breeding associations, decided to focus the program on goats only.

P5, L156: suggest changing "trade" to "import since for the subject of this manuscript, animals leaving the country are not important. Revision accepted.

Regarding transmission between sheep and goats - on the multi-species farms, do the animals lamb and kid together? This may be where transmission occurs. No, in multispecies farms animals do not lamb and kid together. Most of the animals are seasonal breeders and sheep lamb in a different time frame than goats. Even though this was a common practice, breeders understood that keeping kids and lambs together could have been the ideal moment for viral transmission and started keeping them separated.

Round 2

Reviewer 1 Report

The authors only a little improved the manuscript. They did not meet all remarks.

  • First of all the discussion was not improved.
  • Currently, 22 subtypes have been identified within group A. Although the authors wrote "revision accepted" it was not corrected in the manuscript.
  • Since the authors do not present any results regarding genotyping, as this was not their goal, describing the tests for genotyping they used is pointless. I do not understand why these tests were described in such details.

Viruses (as is described in the aim/scope) publish papers that are of significant impact to the virology community. The manuscript does not bring any original/significant  findings to the SRLVs field. I think the work is too weak to be published in such good journal as Viruses

Author Response

Currently, 22 subtypes have been identified within group A. Although the authors wrote "revision accepted" it was not corrected in the manuscript.

We apologize for missing this revision the first time. The number of viral subtypes for genotype A was modified. 

Since the authors do not present any results regarding genotyping, as this was not their goal, describing the tests for genotyping they used is pointless. I do not understand why these tests were described in such details.

The information on the genotyping tests was removed from the materials and methods section.

Some parts of the discussion was modified in order to make it more readable, especially the part discussing the data from two previous pilot studies. 

We thank the reviewer for his/her constructive comments.